# Three New Species of Russulaceae (Russulales, Basidiomycota) from Southern China

**DOI:** 10.3390/jof10010070

**Published:** 2024-01-15

**Authors:** Sen Liu, Mengjia Zhu, Nemat O. Keyhani, Ziyi Wu, Huajun Lv, Zhiang Heng, Ruiya Chen, Yuxiao Dang, Chenjie Yang, Jinhui Chen, Pengyu Lai, Weibin Zhang, Xiayu Guan, Yanbin Huang, Yuxi Chen, Hailan Su, Junzhi Qiu

**Affiliations:** 1State Key Laboratory of Ecological Pest Control for Fujian and Taiwan Crops, College of Life Sciences, Fujian Agriculture and Forestry University, Fuzhou 350002, China; m17633615410@163.com (S.L.); zhumengjia_0529@163.com (M.Z.); 18105011103@163.com (Z.W.); 3328626122@163.com (H.L.); 18965917828@163.com (Z.H.); 19960745421@163.com (R.C.); 15396060715@163.com (Y.D.); cjyang0525@126.com (C.Y.); chenjinhui202312@163.com (J.C.); ndpengyu@163.com (P.L.); 13110661315@163.com (W.Z.); liesleyu@163.com (Y.C.); 2Department of Biological Sciences, University of Illinois, Chicago, IL 60607, USA; keyhani@uic.edu; 3College of Horticulture, Fujian Agriculture and Forestry University, Fuzhou 350002, China; 000q021008@fafu.edu.cn; 4Bureau of Fujian Junzifeng National Nature Reserve, Sanming 365200, China; mxhyb@163.com; 5Institute of Crop Sciences, Fujian Academy of Agricultural Sciences, Fuzhou 350013, China

**Keywords:** Basidiomycota, Russulales, *Russula*, *Lactarius*, new species, morphological and phylogenetic analyses

## Abstract

The characterization of natural fungal diversity impacts our understanding of ecological and evolutionary processes and can lead to novel bioproduct discovery. *Russula* and *Lactarius*, both in the order Russulales, represent two large genera of ectomycorrhizal fungi that include edible as well as toxic varieties. Based on morphological and phylogenetic analyses, including nucleotide sequences of the internal transcribed spacer (ITS), the 28S large subunit of ribosomal RNA (LSU), the second largest subunit of RNA polymerase II (RPB2), the ribosomal mitochondrial small subunit (mtSSU), and the translation elongation factor 1-α (TEF1-α) gene sequences, we here describe and illustrate two new species of *Russula* and one new species of *Lactarius* from southern China. These three new species are: *R. junzifengensis* (R. subsect. Virescentinae), *R. zonatus* (R. subsect. Crassotunicatae), and *L. jianyangensis* (L. subsect. Zonarii).

## 1. Introduction

Fungi classified within the order Russulaceae include members of some of the most significant ectomycorrhizal genera found in almost all forest ecosystems, spanning across temperate, subtropical, and tropical regions [1]. The proliferation and ecological relevance of Russulaceae are evident in the lush forests of southern China and various other regions dispersed throughout South Asia. This fungal assemblage plays a pivotal role in mycorrhizal associations, contributing significantly to the vitality and sustainability of forest ecosystems [2,3]. Russulaceae establish intimate associations with various host plants [4,5]. In addition, these fungi have significant medicinal, nutritional, and bioremediation value, including as resources for novel drug discovery [6,7,8]. Fujian, a coastal province in southern China, is surrounded by mountains on three sides and the sea on the other. The subtropical monsoon climate in this region results in relatively warm, short winters and long, rainy summers compared to northern China [9,10]. Botanically, Fujian is positioned at the southernmost end of the Sino-Japanese Floristic Region and faces Taiwan across the sea, the latter of which belongs to the Indo-Malay Region [11]. The main mountain ranges in Fujian include the Wuyi, Shanling, Jiufeng, and Tailao (elevation of ~200–2158 m), which house subtropical evergreen broadleaf forests, mixed coniferous and broadleaf forests, and South Asian tropical rainforests. The major tree species in Fujian comprise Masson’s pine, bamboo groves, willow trees, banyans, and camphor [12], among which the Chinese fir, Chinese yew, Fujian pine, and Chinese swamp cypress are indigenous to Fujian. The diverse and unique local tree species found in Fujian, coupled with the warm and humid climate, are likely important factors conducive to the proliferation of their associated Russulaceae fungi.

The genus *Russula* Pers. (Russulaceae, Russulales, Basidiomycota) was established by Persoon in 1796 [13]. Members of this genus often constitute crucial components of forest ecosystems via their extensive associations with plants, and also likely as a food source (their fruiting bodies/mushrooms) for a variety of animals [14]. Indeed, a number of *Russula* species are globally recognized as edible fungi [15] and have displayed promising (biopharmaceutical) properties with respect to possessing anticancer and antioxidative activities [16,17]. The morphological classification system for *Russula* is characterized by brightly colored fragile caps, brittle context, amyloid warty spores, abundant sphaerocysts in a heteromerous trama, an absence of latex, and simple-septate hyphae [18,19]. The documented number of species cataloged within the genus *Russula* currently surpasses > 2000, with their fruiting bodies encompassing a vast array of variations in color, morphology, and anatomical characteristics. However, due to the substantial variability exhibited within this taxonomic group, many species still pose considerable challenges in terms of their accurate identification and differentiation. This complexity underscores the likelihood of new species awaiting detection and classification through attempts to compare molecular phylogenetic reconstruction with modern infrageneric classification [20]. Consequently, challenges persist in differentiating and taxonomically categorizing *Russula* species within fungal surveys and ethnopharmacological investigations [21]. 

The genus *Lactarius* also belongs to the family Russulaceae in the order Russulales [22]. In traditional classification, all species that exude latex (or “milk”, hence the term “milkcap” fungi) were grouped under the genus *Lactarius*. Buyck et al. [22,23] separated *Lactifluus* and *L. furcatus* Coker from the genus *Lactarius*, establishing a new genus named *Lactarius* sensu novo, which is mainly classified into three subgenera: *L.* subg. *Lactarius*, *L.* subg. *Plinthogalus* (Burl.) Hesler & A.H. Sm., and *L.* subg. *Russularia* (Fr.) Kauffman. Although the exploration of extensive fungal resources has led to the identification of several dozen Russulaceae species across various regions [24,25,26,27], investigations concerning this genus in southern China remain inadequately addressed, with the continual discovery of new species [28,29].

During an exploration aimed at delineating the diversity and geographical distribution of *Russula* in China, a series of intriguing samples was gathered within Fujian province, China. These isolates displayed characteristics that did not correspond to any known species within the genus. Employing both morphological and molecular phylogenetic analyses, we identify three new species within the Russulaceae family. We present detailed descriptions of these newfound species, complemented by illustrations elucidating their distinctive morphological attributes. 

## 2. Materials and Methods

### 2.1. Collections and Morphological Analyses

Fresh fruiting bodies of two unknown (putative members of the *Russula*) mushrooms were collected from the Junzifeng National Nature Reserve, and one from Jiufeng Mountain, Jianyang (putative member of the *Lactarius*), in the Fujian Province, China, in August 2021. These specimens were collected during field expeditions focused on fungi. Images of the fresh fruiting bodies were captured using a Canon (Tokyo, Japan) EOS 6D Mark II camera. The meticulous documentation of their macroscopic attributes involved the careful examination of fresh samples in their natural diurnal environment. Comprehensive records encompassing macroscopic characteristics and habitat specifics were meticulously collated from collection records and accompanying visual documentation, adhering to the conventions of mycological taxonomic research. To ensure the permanent preservation of specimens, one crucial step was a dehydration process, during which the specimens underwent desiccation within a drying oven set at 45 °C. This meticulous procedure persisted until the moisture content of the fruiting bodies was diligently reduced to below 10%, ensuring their suitability for long-term storage. Microsections of dried specimens were stained with a mixture of 5% potassium hydroxide (KOH) and 1% Congo red. A detailed illustration of the structure and ornamentation of the spores was carried out using a scanning electron microscope (ZEM15C, ZEPTOOLS, Tongling, China). Microscopic features were observed using a Leica microscope (DM2500, Wetzlar, Germany) at magnifications up to 100×. For the description of basidiospores, 20 basidiospores, in profile view, were measured. The basidium length excludes sterigmata. The notation (a-)b–c(-d) was used to represent the dimensions of basidiospores, where the range ‘b–c’ covered 90% or more of the measured values. ‘a’ and ‘d’ represent the extreme values. An average length/width ratio (Q value) was calculated from 20 spores, along with the standard deviation, reflecting the characteristics of the basidiospores. The specimens were deposited in the Herbarium Mycologicum Academiae Sinicae (HMAS) at the Institute of Microbiology, Chinese Academy of Sciences, with the specimen numbers HMAS 298099, HMAS 298100, HMAS 298101, HMAS 298102, HMAS 298103, and HMAS 298104. Taxonomic information on the new taxa was submitted to MycoBank (http://www.mycobank.org (accessed on 10 January 2024)).

### 2.2. DNA Extraction, PCR Amplification, and Sequencing

DNA was extracted using the Fungal DNA Mini Kit (OMEGA-D3390, Feiyang Biological Engineering Corporation, Guangzhou, China) following the manufacturer’s protocol. Briefly, 100 mg of starting material (fruiting body) was used for DNA extraction. The amplification of the nucleotide sequences of the internal transcribed spacer (ITS), 28S large subunit regions of ribosomal DNA (LSU), the translation elongation factor 1-α (TEF1-α), the ribosomal mitochondrial small subunit (mtSSU), and second largest RNA polymerase II regions (RPB2) was conducted via polymerase chain reaction (PCR) using the primer pairs: ITS4/ITS5 [30], LROR/LR5 [31], TEF1-α [32], mtSSU [30], and RPB2-6F/RPB2-7cR [33], respectively. The PCR reaction volume was 25 μL, comprising 12.5 μL of 2× Rapid Taq Master Mix (Vazyme, Nanjing, China), 1 μL of each forward and reverse primer (10 μM) (Sangon, Shanghai, China), and 1 μL of template genomic DNA. The reaction mixture was adjusted to a total volume of 25 μL using distilled deionized water. Amplification products were visualized using 1% agarose gel electrophoresis. Sequencing was performed by Fuzhou Tsingke Company (Fuzhou, China) using bidirectional (double-stranded) sequencing. 

### 2.3. Alignment and Phylogenetic Analyses

To construct the phylogenetic tree of Russulaceae, we utilized sequences obtained from six fungal strains and reference sequences for multi-locus phylogenetic analyses which were obtained from Rehner and Buckley [32], Chen et al. [34], Deng et al. [35], Buyck et al. [36], and Roy et al. [37]. The newly generated sequences were screened for similarity through a GenBank BLAST search. The ITS, LSU, mtSSU, RPB2, and TEF1-α sequences were aligned using the MAFFT v. 7.11 online tool (https://mafft.cbrc.jp/alignment/software/ (accessed on 23 December 2023)), followed by manual adjustments in MEGA 7.0. Phylogenetic analyses employed both maximum likelihood (ML) and Bayesian inference (BI) methods. ML analysis was conducted using RaxML-HPC2 on XSEDE v. 8.2.12 via the CIPRES Science Gateway portal, while BI analysis was performed using MrBayes on XSEDE v. 3.2.7a (https://www.phylo.org/ (accessed on 25 December 2023)). The consensus tree was constructed using FigTree v. 1.4.4 and further refined using Adobe Illustrator CS 6.0. Newly generated sequences from this study have been deposited in GenBank. Branches showing ML bootstrap support values (≥70) and Bayesian posterior probability (≥0.90) were considered significantly supported.

## 3. Results

### 3.1. Phylogenetic Analyses

The multi-locus sequence matrix spans a length of 4350 bp. Its dataset comprises 700 bp of ITS, 890 bp of LSU, 1100 bp of TEF1-α, 800 bp of RPB2, and 860 bp of mtSSU. For the multi-locus region, the best substitution model for ITS and RPB2 in the BI analysis is SYM + G4, while for LSU and TEF1-α it is SYM + I + G4, and for mtSSU the best substitution model is GTR + F + G4. A total of 148 sequences, including newly generated ones, were deposited in the GenBank database (Table 1 and Table 2). Based on the foundational rank consistency of the phylogenetic topologies obtained from BI and ML analyses, only the ML trees are depicted in Figure 1 and Figure 2. The resulting phylogenetic trees demonstrate strong support for clades of the four new species in multi-locus phylogenetic analyses. These new species exhibit notable distinctions from known species (Figure 1 and Figure 2). Bootstrap and posterior probability values indicate robust support in multi-locus phylogeny for *R. junzifengensis* (from subsect. Virescentinae), *R. zonatus* (from subsect. Virescentinae), and *L. jianyangensis* (L. subsect. Zonarii), forming a distinct clade.

*Lactarius jianyangensis* showed the greatest similarity to *Lactarius pallido-ochraceus*, with an additional 84 sequences from *Lactarius* collected to construct the tree (Table 1, Figure 1). *L. jianyangensis* exhibited the highest genetic similarity to *L. pallido-ochraceus* and clustered with two other species, *L. vulgaris* and *L. pallidizonatus* (Figure 1). However, despite clustering with these three species, the substantial phylogenetic distance between *L. jianyangensis* and the other members of this clade supports its classification as an independent species. Two putative new species within the *Russula* genus, *Russula junzifengensis*, formed a strongly supported cluster (BS 100%) and were notably distinct from other known species within the Virescentinae group. *Russula junzifengensis* clustered together with an unidentified sequence from China (voucher: HMAS250919), which served as the sister clade to *R. indoalba*, supported by 98% bootstrap support and a posterior probability of 1. *R. zonatus* clustered alongside *R. brunneoaurantiaca* and formed a clade sister to *R. brunneoaurantiaca* with a posterior probability of 1.

### 3.2. Taxonomy

*Russula zonatus* S. Liu & Jun Z. Qiu, sp. nov. (Figure 3a,b and Figure 4).

MycoBank: MB 851147. 

Etymology: The epithet “*zonatus*” refers to the morphological feature of ring patterns on the surface.

Holotype: CHINA. Fujian Province, Mingxi County, Xiayang Town, Ziyun Village, in mixed forests, alt. 379 m, 26.34323138° N, 117.45805533° E, 30 August 2021, S. Liu and Jun Z. Qiu (holotype HMAS298100; paratype HMAS298099).

Description: Basidiocarps medium-sized to big. Pileus 7.5–11.9 cm in diam., convex-expanded to infundibuliform with a central depression and slight incurved margin, shallowly infundibuliform when mature, surface glabrous and dry with unclear or golden brown zone lines, no dark brown ring patterns on the surface, brown to grayish brown. Context 3 mm thick, satin white. Lamellae adnate, crowded, marble white, no forking, concolorous with the pileus when fully mature. Stipe 2.5 × 6–3.5 × 10 cm, central, equal, sometimes with fibrils, whitish or sub-concolorous with the pileus.

Basidiospores (6.4) 6.5–7.8 (8) × (5.2) 5.5–6.8 (7) µm (Q = (1.05) 1.06–1.19 (1.23), Q = 1.13 ± 0.05), broadly ellipsoid, with almost isolated warts, plage not amyloid. Basidia 32.5–49 × 10–13 µm, clavate, four-spored. Cystidium common, 30–50 × 5–7.5 µm, fusiform to lanceolate with campulitropal head. Lamella edge sterile, lamellae 8.74 µm thick.

Pileipellis duplex: gelatinous epicutis, 100–230 µm thick, hyphae 2–7 µm wide, hyaline to light yellow in 5% KOH, smooth. Hypodermium well developed, yellowish to lightly yellow intracellular pigment in 5% KOH.

Ecology and distribution: Gregarious in subtropical mixed forests (fagaceous forests or mixed forests with fagaceous trees). Known to inhabit Fujian Province, China.

*Russula junzifengensis* S. Liu & Jun Z. Qiu, sp. nov. (Figure 3c,d and Figure 5).

MycoBank: MB 851146. 

Etymology: Named after the Junzifeng Nature Reserve where the fungus was collected.

Holotype: CHINA. Fujian Province, Mingxi County, Junzifeng Nature Reserve, Xiafang Town, Zhushe Village, in mixed forests, alt. 410 m, 26.56490316° N, 117.03322856° E, 7 August 2021, S. Liu and Jun Z. Qiu (holotype HMAS298101; paratype HMAS298102).

Description: Basidiomata medium-sized, with a diameter of 40–60 mm. Initially hemispherical, later broadly convex to flat with a shallow depression, featuring a sub-transparently striate margin. The lamellae are densely crowded and sometimes slightly decurrent in mature and dry conditions, with a sharp, incurved, and even margin. The surface is glabrous, ranging from dry to slightly glutinous, presenting a satin white appearance, marble white at the center. In the mature stage, the central color turns to a shade of light yellow, pinard yellow, occasionally displaying maize yellow or light orange–yellow to capucine buff. The lamellae are adnate, densely packed, and of yellowish white color, without forking, becoming fragile and matching the pileus color when fully mature. The stipe measures 5.5 × 1.2 cm, central, cylindrical to slightly tapered upwards, rarely becoming subcylindrical to clavate, slightly narrowing towards the base, without an annulus. The stipe is white, appearing yellowish white, smooth in youth, later exhibiting fibrils on the surface. While young, it is full-bodied, eventually becoming hollow. The odor is indistinct.

Basidiospores (5.4) 6–7.5 (7.5) × (5.2) 5.5–6.5 (6.6) µm, [Q = (1.04) 1.05–1.25 (1.25), Q = 1.15 ± 0.06], ellipsoid, composed of small amyloid conical warts, mostly isolated, sometimes fused; indextrinoid, ornamentation small to medium-sized. Basidia 38.5–46.5 × 9–12 µm (smaller), clavate, four-spored, clavate, hyaline to light yellow in 5% KOH. Cystidia, 42 ×13.5 µm (smaller), thin-walled, hyaline in 5% KOH. Pileipellis well developed, separated from the surrounding spherocytes of the context, yellowish to lightly yellow intracellular pigment in 5% KOH.

Ecology and distribution: Gregarious in subtropical mixed forests (solitary or gregarious in Fagaceae forest). Known to inhabit Fujian Province, China.

*Lactarius jianyangensis* S. Liu & Jun Z. Qiu, sp. nov. (Figure 3e,f and Figure 6).

MycoBank: MB 851149.

Etymology: Named after Jianyang District, where the fungus was collected.

Holotype: CHINA. Fujian Province, Nanping City, Jianyang District, in mixed forests, alt 841 m, 27.34220872° N, 118.16609715° E, 19 August 2021, S. Liu and Jun Z. Qiu (holotype HMAS298103; paratype HMAS298104).

Description: Basidiomata with a small size. Pileus 25–45 mm broad, initially hemispheric, becoming plano-convex and planate when mature, convex with inrolled margin, shallowly infundibuliform when mature, surface greasy when wet, aniline yellow, bittersweet pink, Titian red to agate, sometimes center salmon-orange or Mars yellow with a raw sienna margin, margin glabrous, sub-transparently striate. Context 3–4 mm thick, whitish to brown. The lamella 1–2 mm broad, Mikado orange to cadmium orange when young, xanthine orange, amber brown when mature, concolorous with the pileus when fully mature, sub-crowded to crowded, unequal length and extended. Additionally, the flesh of *Lactarius* has an aroma. Stipe 35–40 × 7–10 mm, central or tapering downwards, sometimes with longitudinal grooves, surface smooth, greasy, with scattered pits, whitish or sub-concolorous with the pileus, the end of the stipe was slightly enlarged, succulent and hollow, latex white or watery–milky.

Basidiospores (5.99) 6–7.7 (7.8) × (4.8) 4.9–6.3 (6.44) μm, [Q = (1.07) 1.09–1.34 (1.36), Q = 1.20 ± 0.09], broadly ellipsoid, surface has protuberance ridge to reticulate pattern, colorless to hyaline in KOH. Basidia 30.55–39.03 × 8.21–11.32 μm, four-spored, narrowly clavate, colorless to hyaline in KOH, sterigmata 2.84–3.87 μm. Clamp connections abundant in all tissues. The head of pleuromacrocystidia is warping, trama 4.53–4.81 μm.

Ecology and distribution: Gregarious in subtropical fagaceous forests. Known to inhabit Fujian Province, China.

## 4. Discussion

All *Russula* and *Lactarius* species characterized thus far form ectomycorrhizal symbioses with higher plants and trees, and both genera contain cosmopolitan as well as more host-specific members, with both edible and toxic species having been identified [20,38]. *Lactarius* is characterized by the production of latex, although the genus has now been separated into two (*Lactarius* and *Lactifluus*), with an additional separation of several species from *Lactarius* as well as *Russula* into *Multifurca* [39]. Most species of Lactarius form symbioses with broadleaf or coniferous hosts, consistent with their discovery in the Fujian forests of pine.

Russula are distinguished by their bright-colored caps, but do not produce latex and are often characterized by their brittle caps [40]. Due to the difficulty in separating species by their morphological characteristics alone, the modern identification of species within the *Russula* and *Lactarius* genera has relied on utilizing the ITS sequence in phylogenetic analysis as the primary molecular method for distinguishing and interpreting these closely related species [41]. However, an overreliance on ITS-based phylogenetic structures can lead to inaccurate subgenus classifications and may overlook the presence of known species, such as the *R. queletii* complex and the rhodochroa-subsanguinaria complex, often manifested within ITS-based phylogenetics [24]. Additionally, earlier studies focusing on *Lactarius* species found minimal consistency between Asian *Lactarius* species and those from other continents. Relying solely on ITS-based phylogenetic analysis and morphological characteristics for *Lactarius* species’ identification appears insufficient. Hence, the utilization of multi-locus phylogenetic analysis has become the preferred method for revealing the genetic relationships within the *Russula* and *Lactarius* genera.

Here, we employed a combined ITS-nrLSU-RBP2-mtSSU-TEF-1α multi-locus phylogenetic analysis method to support the identification of three species that have been named *R. junzifengensis*, *R. zonatus*, and *L. jianyangensis*. These assignments are based on combined morphological characterizations and molecular multilocus phylogenetic analyses. *Russula zonatus* appears to be a very common red mushroom in the subtropical-tropical Quercus forests of Fujian and is a member of the subgenus Crassotunicata. Key features for its identification include medium basidiocarps, a convex expanded to infundibuliform pileus with a central depression and slight incurved margin, glabrous and dry surface with indistinct or golden brown zone lines, brown to grayish brown color, very crowded lamellae, moderately ornamented basidiospores with isolated warts, and a subtropical habitat. *R. zonatus* forms a clade with *R. brunneoaurantiaca* (with a highest ITS identity of ~99%)*, R. adusta*, and *R. nigricans*. All these species have a mucilaginous pileus and comparatively large spores and basidia. *R. zonatus* has a high similarity to *R.brunneoaurantiaca*’s ITS sequence (0.99%), but they have significant morphological differences. *R. zonatus* belongs to the subgenus Crassotunicata of the *Russula* genus and has medium to large basidiocarps, with the convex expanded to the infundibuliform pileus with a central depression and slight incurved margin. It has a glabrous and dry surface with indistinct or golden brown zone lines, a brown to grayish brown color, very crowded lamellae, and moderately ornamented basidiospores with isolated warts. *R. zonatus* has smaller basidia and cystidia compared to *R. brunneoaurantiaca*, and, at the macroscopic scale, additional differences are quite obvious, with *R. brunneoaurantiaca* having a surface that is mucilaginous, brownish orange turning yellowish brown to light brown, and a smooth stipe surface [37].

*Russula junzifengensis* is characterized by a white or slightly stained white–yellow or yellow pileus, which is broadly convex to flat with a shallow depression, slightly crowded lamellae, medium basidiospores with isolated warts, and a subtropical habitat. This species is similar to *R. pseudocrustosa*, *R. indoalba*, and *R. xanthovirens*, and phylogenetic analysis showed that *R. junzifengensis* formed a highly supported sister group with *R. indoalba*, but their ITS sequence similarity is less than 90%. Macromorphologically this species seems to be indistinguishable from *R. indoalba*; both species have whitish gray basidiomata, a clavate stipe, and ellipsoid basidiospores. But the lamellae of *R. junzifengensis* are not attached to the stipe and appear with fibrils on the stipe.

The phylogenetic results indicate that *L. jianyangensis* is closest to *L. pallido-ochraceus* and *L. pallidizonatus*. In comparison to *L. pallido-ochraceus*, both species have few pits on their stipe, a surface that is greasy when wet and basidiospores with reticulate ornamentation. In comparison to *L. pallido-ochraceus*, the basidiomata of the new species described here has a deeper color, smaller basidia, and narrower pleuromacrocystidia. The ITS similarity between the two species was 94%. Southern China’s *Lactarius pallidizonatus* X.H. Wang seems to be another closely related species [39]; *L. pallidizonatus* can be distinguished by its light orange–yellow margin, cream–whitish context, pits near the base on stipes, and basidiospores (80/4/3) (7.0) 7.5–9.0 (9.5) × (5.5) 6.0–7.5 µm [Q = (1.10) 1.12–1.29 (1.35), Q = 1.21 ± 0.05], however, *L. pallidizonatus* has a lighter color and bigger basidiospores than *L. jianyangensis*.

## 5. Conclusions

In this study, we employed a multi-locus phylogenetic analysis method, combined with morphological characteristics, to identify three new fungal species in the Quercus forests of Fujian province, namely *R. junzifengensis*, *R. zonatus*, and *L. jianyangensis*. These new species belong to the *Russula* and *Lactarius* genera, which form ectomycorrhizal symbioses with higher plants and trees. We found that relying solely on ITS sequences and morphological characteristics for species identification is insufficient, as it may lead to inaccurate subgenus classifications and the omission of known species. Therefore, we suggest using multi-locus phylogenetic analysis methods to reveal the genetic relationships within the *Russula* and *Lactarius* genera, as well as their phylogenetic affinities with species from other geographical regions. Our study provides new data on the fungal diversity and distribution in Fujian province, and also contributes new insights to fungal taxonomy and phylogeny.

## Figures and Tables

**Figure 1 jof-10-00070-f001:**
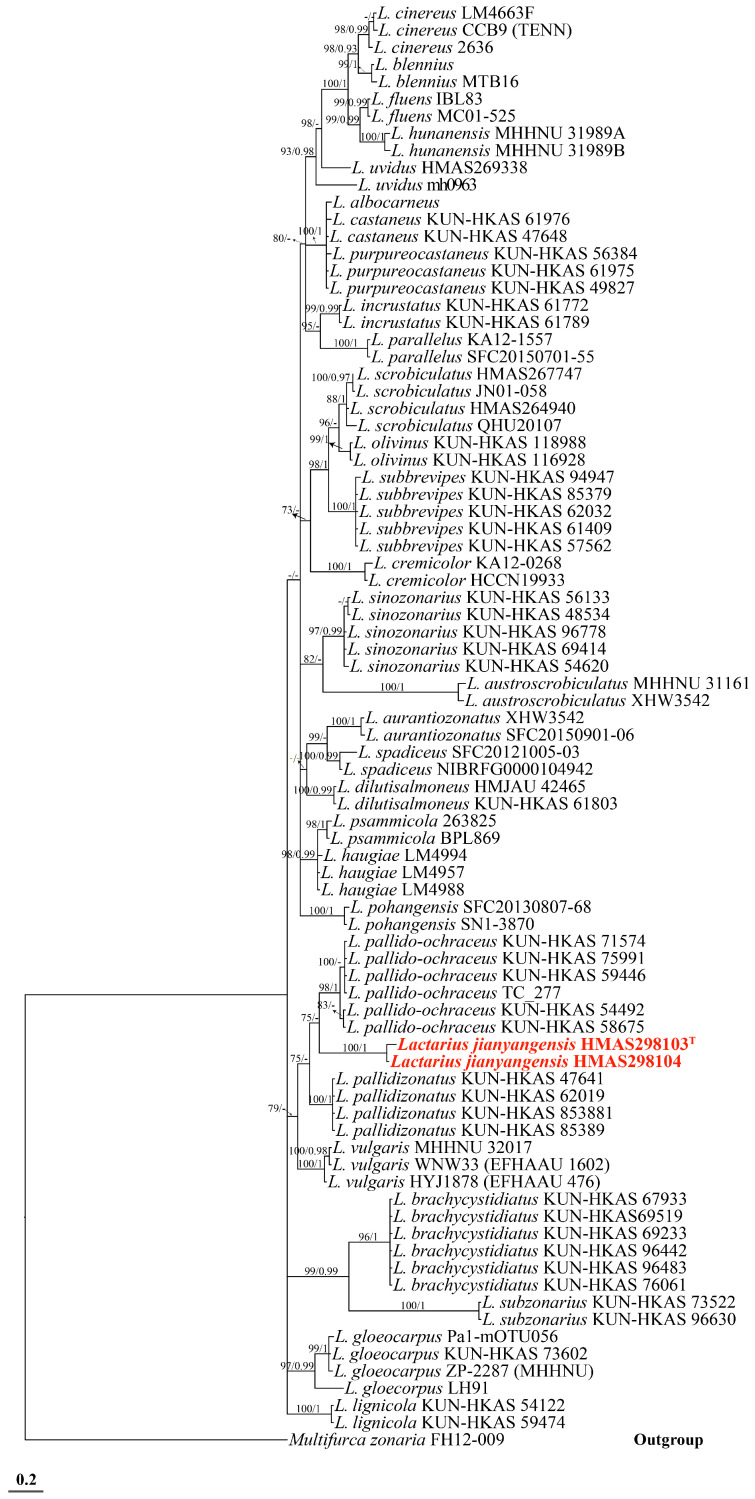
Phylogeny inferred from *Lactarius* multigene sequences (nrLSU, ITS, mtSSU, rpb2, and tef1-α) using Bayesian analysis. Support values in normal type are bootstrap support (BS, significant when ≥70%). Values in bold are Bayesian Posterior Probabilities (PP, significant when ≥0.95). The scale bar indicates the number of nucleotide substitutions per site. New species are highlighted in red. Arrows show the support values at the branching points. Superscript “T” denotes the type strain of the new species.

**Figure 2 jof-10-00070-f002:**
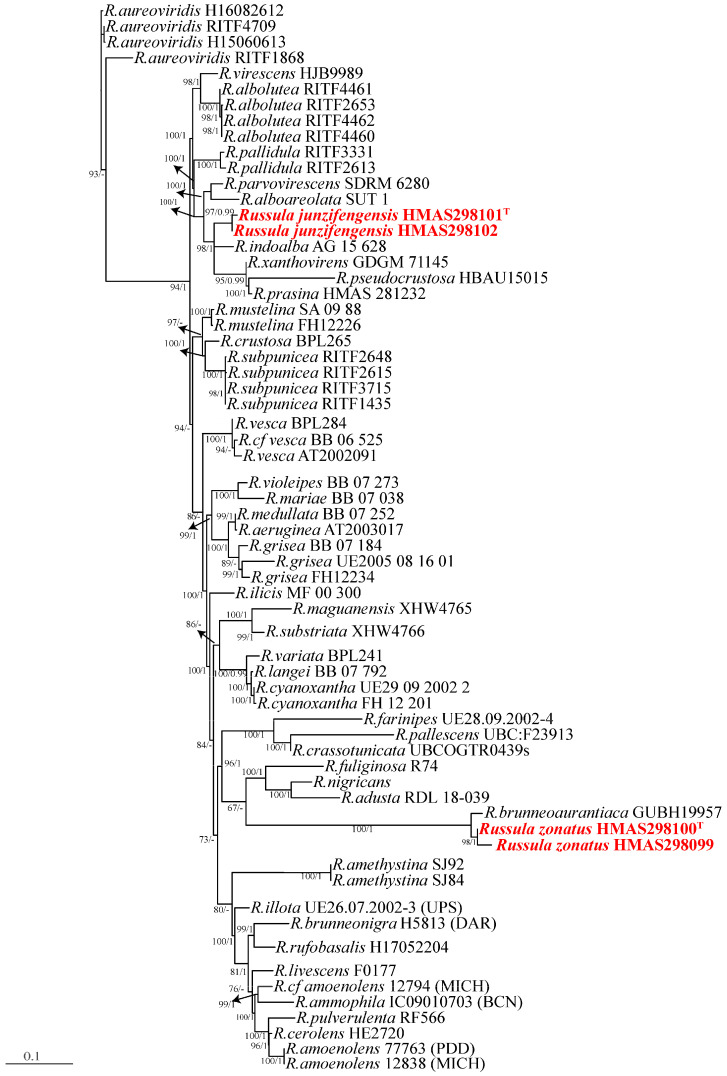
Phylogeny inferred from *Russula* multigene sequences (nrLSU, ITS, mtSSU, rpb2, and tef1-α) using Bayesian analysis. Support values in normal type are bootstrap support (BS, significant when ≥70%). Values in bold are Bayesian Posterior Probabilities (PP, significant when ≥0.95). The scale bar indicates the number of nucleotide substitutions per site. New species are highlighted in red. Arrows show the support values at the branching points. Superscript “T” denotes the type strain of the new species.

**Figure 3 jof-10-00070-f003:**
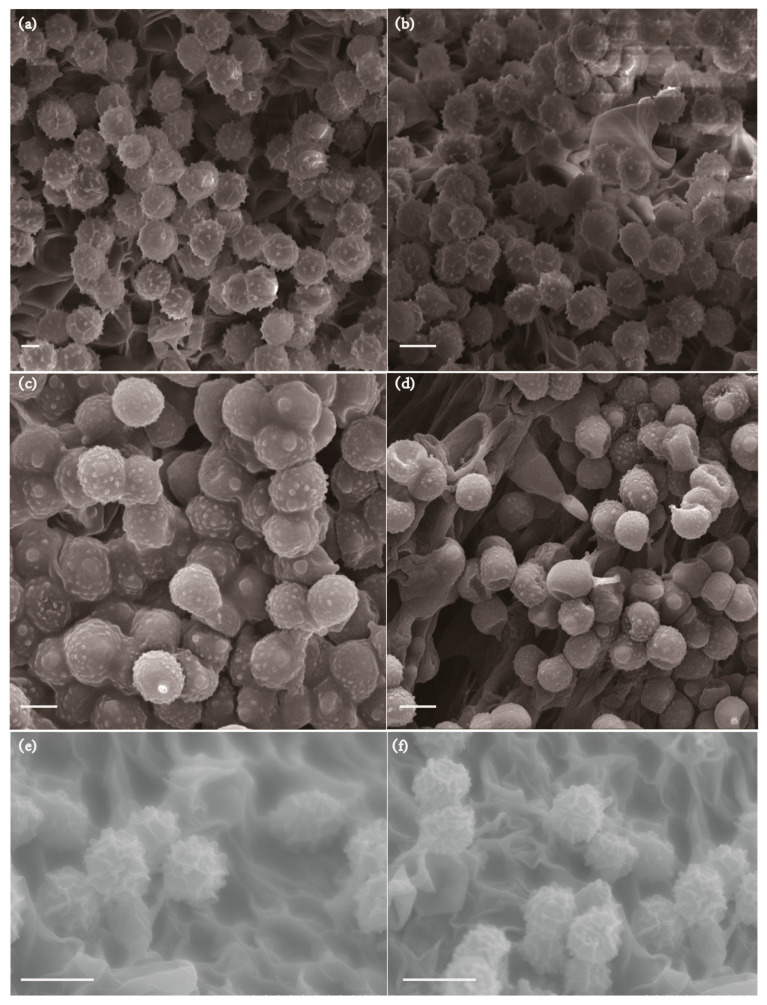
SEM photos of basidiospores. (**a**,**b**) *R. zonatus*, (**c**,**d**) *R. junzifengensis*, and (**e**,**f**) *L. jianyangensis*. Scale bars: (**a**–**f**) = 10 μm.

**Figure 4 jof-10-00070-f004:**
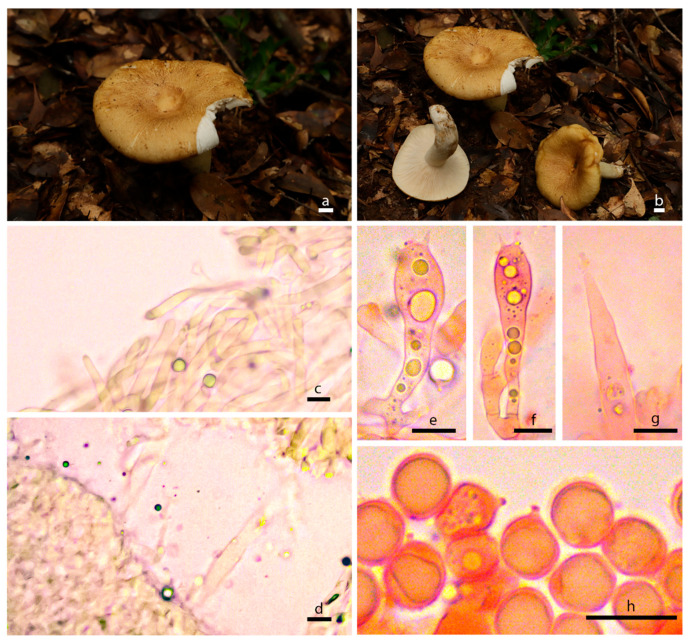
Morphological characteristics of *Russula zonatus* (HMAS298100). (**a**,**b**) Basidiomata; (**c**) pileipellis in 5% KOH; (**d**) lamellae in 5% KOH. (**e**,**f**) Basidium in Congo Red reagent; (**g**) cystidium in Congo Red reagent; (**h**) basidiospores in Congo Red reagent; bars: (**a**,**b**) = 1 cm; (**c**–**h**) = 10 µm.

**Figure 5 jof-10-00070-f005:**
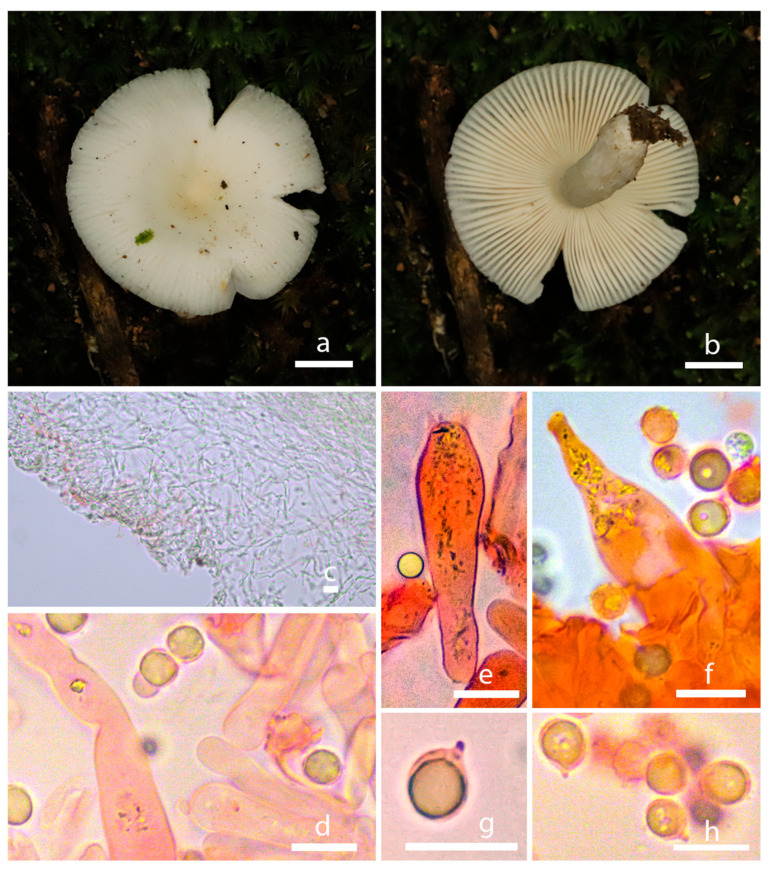
Morphological characteristics of *Russula junzifengensis* (HMAS298101). (**a**,**b**) Basidiomata; (**c**) pileipellis in 5% KOH; (**d**) lamellae in Congo Red reagent. (**e**) Basidium in Congo Red reagent; (**f**) cystidium in Congo Red reagent; (**g**,**h**) basidiospores in Congo Red reagent; bars: (**a**,**b**) = 1 cm; (**c**–**h**) = 10 µm.

**Figure 6 jof-10-00070-f006:**
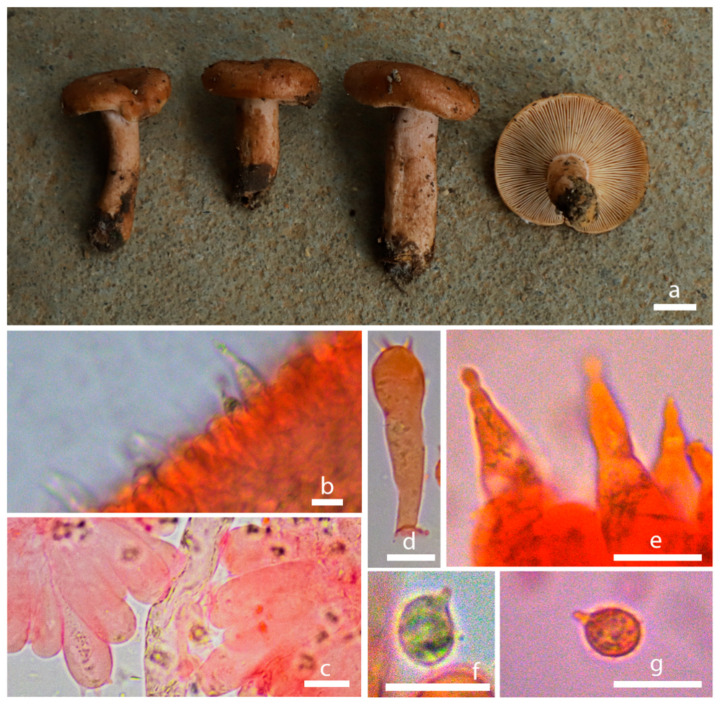
Morphological characteristics of *Lactarius jianyangensis* (HMAS298103). (**a**) Basidiomata; (**b**) pileipellis in Congo Red reagent (**c**) lamellae in Congo Red reagent. (**d**) Basidium in Congo red reagent; (**e**) cystidium in Congo red reagent; (**f**,**g**) basidiospores in Congo red reagent; bars: (**a**) = 1 cm; (**b**–**g**) = 10 µm.

**Table 1 jof-10-00070-t001:** Species and specimens of *Lactarius* used for the molecular phylogenetic analyses.

Taxon	Voucher	Location	GenBank Accession Number			
			ITS	nrLSU	RPB2	TEF1
*L. albocarneus*	-	China	KX441117	KX441364	KX442105	-
*L. aurantiozonatus*	HCCN10589	Korea	MH984993	MH985118	MH936950	-
*L. aurantiozonatus*	SFC20150901-06	Korea	MH984976	MH985097	MH936929	-
*L. austroscrobiculatus*	MHHNU 31161	China	OL770185	-	-	-
*L. austroscrobiculatus*	XHW3542	China	OL770183	-	-	-
*L. blennius*	MTB16	Germany	MN947353	-	-	-
*L. blennius*	-	Sweden	AY606944	-	-	-
*L. brachycystidiatus*	KUN-HKAS 67933	China	MF508951	-	-	-
*L. brachycystidiatus*	KUN-HKAS 96483	China	MF508950	-	-	-
*L. brachycystidiatus*	KUN-HKAS 96442	China	MF508949	-	-	-
*L. brachycystidiatus*	KUN-HKAS 69233	China	MF508948	-	-	-
*L. brachycystidiatus*	KUN-HKAS 76061	China	MF508947	-	-	-
*L. brachycystidiatus*	KUN-HKAS69519	China	MF508952	-	-	-
*L. castaneus*	KUN-HKAS 47648	China	MF508962	-	-	-
*L. castaneus*	KUN-HKAS 61976	China	MF508961	-	-	-
*L. cinereus*	LM4663F	Mexico	FJ348708	-	-	-
*L. cinereus*	CCB9 (TENN)	USA	MF755272	-	-	-
*L. cinereus*	2636	Canada	KJ705204	-	-	-
*L. cremicolor*	KA12-0268	Korea	MH985013	-	QCH40198	-
*L. cremicolor*	HCCN19933	Korea	MH984972	-	QCH40153	-
*L. dilutisalmoneus*	HMJAU 42465	China	MF152847	-	-	-
*L. dilutisalmoneus*	KUN-HKAS 61803	China	MF152846	-	-	-
*L. fluens*	IBL83	Poland	MZ410712	-	-	-
*L. fluens*	MC01-525	Denmark	AJ889961	-	-	-
*L. gloecorpus*	LH91	USA	GQ268638	-	-	-
*L. gloeocarpus*	KUN-HKAS 73602	China	OL770166	-	-	-
*L. gloeocarpus*	ZP-2287 (MHHNU)	China	OL770165	-	-	-
*L. gloeocarpus*	Pa1-mOTU056	Japan	LC315865	-	-	-
*L. haugiae*	LM4994	Mexico	KT583642	-	AOF41440	-
*L. haugiae*	LM4988	Mexico	KT583641	-	AOF41439	-
*L. haugiae*	LM4957	Mexico	KT583640	-	-	-
*L. hunanensis*	MHHNU 31989B	China	OL770172	-	-	-
*L. hunanensis*	MHHNU 31989A	China	OL770171	-	-	-
*L. incrustatus*	KUN-HKAS 61789	China	MK675285	-	-	-
*L. incrustatus*	KUN-HKAS 61772	China	MK675284	-	-	-
*L. jianyangensis*	HMAS298103^T^	China	OR835448	OR826782	OR915862	OR887738
*L. jianyangensis*	HMAS298104	China	OR835446	PP033514	OR915863	OR887739
*L. lignicola*	KUN-HKAS 59474	China	MF508946	-	-	-
*L. lignicola*	KUN-HKAS 54122	China	MF508945	-	-	-
*L. olivinus*	KUN-HKAS 116928	China	OL770196	-	-	-
*L. olivinus*	KUN-HKAS 118988	China	OL770195	-	-	-
*L. pallidizonatus*	KUN-HKAS 85389	China	MF508932	-	-	-
*L. pallidizonatus*	KUN-HKAS 62019	China	MF508931	-	-	-
*L. pallidizonatus*	KUN-HKAS 85388	China	MF508933	-	-	-
*L. pallidizonatus*	KUN-HKAS 47641	China	MF508930	-	-	-
*L. pallido-ochraceus*	KUN-HKAS 71574	China	MF508942	-	-	-
*L. pallido-ochraceus*	KUN-HKAS 54492	China	MF508941	-	-	-
*L. pallido-ochraceus*	KUN-HKAS 58675	China	MF508940	-	-	-
*L. pallido-ochraceus*	TC_277	China	MW722813	-	-	-
*L. pallido-ochraceus*	KUN-HKAS 59446	China	MF508943	-	-	-
*L. pallido-ochraceus*	KUN-HKAS 75991	China	MF508939	-	-	-
*L. parallelus*	SFC20150701-55	Korea	MH984953	MH985072	MH936904	-
*L. parallelus*	KA12-1557	Korea	MH984921	MH985035	MH936867	-
*L. pohangensis*	SFC20130807-68	Korea	MH985018	MH985143	MH936975	-
*L. pohangensis*	SN1-3870	China	LC622651	-	-	-
*L. psammicola*	263825	USA	MK607513	-	-	-
*L. psammicola*	BPL869	USA	KY848507	-	-	-
*L. purpureocastaneus*	KUN-HKAS 61975	China	MF508965	-	-	-
*L. purpureocastaneus*	KUN-HKAS 56384	China	MF508964	-	-	-
*L. purpureocastaneus*	KUN-HKAS 49827	China	MF508963	-	-	-
*L. scrobiculatus*	QHU20107	China	OM970920	-	-	-
*L. scrobiculatus*	HMAS264940	China	KX441085	KX441332	KX442073	-
*L. scrobiculatus*	JN01-058	Thailand	KF432968	-	-	-
*L. scrobiculatus*	HMAS267747	China	KX441098			MF893430
*L. sinozonarius*	KUN-HKAS 69414	China	MF508926	-	-	-
*L. sinozonarius*	KUN-HKAS 56133	China	MF508929	-	-	-
*L. sinozonarius*	KUN-HKAS 48534	China	MF508928	-	0	-
*L. sinozonarius*	KUN-HKAS 96778	China	MF508927	-	-	-
*L. sinozonarius*	KUN-HKAS 54620	China	MF508925	-	-	-
*L. spadiceus*	SFC20121005-03	Korea	MH985021	MH985146	MH936978	-
*L. spadiceus*	NIBRFG0000104942	Korea	MH984956	MH985076	MH936908	-
*L. subbrevipes*	KUN-HKAS 85379	China	MF508938	-	-	-
*L. subbrevipes*	KUN-HKAS 61409	China	MF508937	-	-	-
*L. subbrevipes*	KUN-HKAS 57562	China	MF508936	-	-	-
*L. subbrevipes*	KUN-HKAS 62032	China	MF508935	-	-	-
*L. subbrevipes*	KUN-HKAS 94947	China	MF508934	-	-	-
*L. subzonarius*	KUN-HKAS 96630	China	MF508960	-	-	-
*L. subzonarius*	KUN-HKAS 73522	China	MF508959	-	-	-
*L. uvidus*	mh0963	Sweden	AY606957	AF325293	-	-
*L. uvidus*	HMAS269338	China	KX441140	KX441387	KX442128	-
*L. vulgaris*	MHHNU 32017	China	OL770178	-	-	-
*L. vulgaris*	HYJ1878 (EFHAAU 476)	China	OL770175	-	-	-
*L. vulgaris*	WNW33 (EFHAAU 1602)	China	OL770173	-	-	-
*Multifurca zonaria*	FH12-009	Thailand	KF432960	-	-	-

Superscript “T” denotes the type strain of the new species.

**Table 2 jof-10-00070-t002:** Species and specimens of *Russula* used for the molecular phylogenetic analyses.

Taxon	Voucher	Location	GenBank Accession Number				
			ITS	nrLSU	RPB2	mtSSU	TEF1
*R. adusta*	RDL 18-039	Belgium	OM833079	-	-	-	ON015965.1
*R. aeruginea*	AT2003017	Sweden	DQ421999	DQ421999	-	-	-
*R. alboareolata*	SUT-1	Thailand	AF345247	-	-	-	-
*R. albolutea*	RITF4460	China: Chongqing	-	MW397121	MW411341	MW403834	-
*R. albolutea*	RITF4461	China: Yunnan	-	MW397122	MW411342	MW403835	-
*R. albolutea*	RITF4462	China: Yunnan	-	MW397123	MW411343	MW403836	-
*R. albolutea*	RITF2653	China	MT672478	MW397120	MW411340	MW403833	-
*R. amethystina*	SJ84	Pakistan	KT953615	-	-	-	-
*R. amethystina*	SJ92	Pakistan	KT953616	-	-	-	-
*R. amoenolens*	12838 (MICH)	France	KF245510	-	-	-	-
*R. amoenolens*	77763 (PDD)	New Zealand	GU222264	-	-	-	-
*R. ammophila*	IC09010703 (BCN)	Spain	MK112566	MK108033	-	-	-
*R. aureoviridis*	H15060613	China	KY767808	-	-	-	-
*R. aureoviridis*	RITF1868	China	MW397096	-	-	MW403842	-
*R. aureoviridis*	H16082612	China	KY767809	MK881920	-	MK882048	MN617846
*R. aureoviridis*	RITF4709	China	MW646980	MW646992	-	MW647003	MW650849
*R. brunneoaurantiaca*	GUBH19957	India	OP270714	-	-	-	-
*R. brunneonigra*	H5813 (DAR)	Australia	EU019945	-	-	-	-
*R. cerolens*	HE2720	China	KC505578	-	-	-	-
*R. cf. amoenolens*	12794 (MICH)	USA	KF245512	-	-	-	-
*R. cf*. *vesca*	BB 06.525	Mexico	-	KU237465	KU237751	KU237309	-
*R. crassotunicata*	UBCOGTR0439s	Canada	EU597082	-	-	-	-
*R. crustosa*	BPL265	USA: Tennessee	-	KT933826	KT933898	-	-
*R. cyanoxantha*	UE29.09.2002-2	France	-	DQ422033	DQ421970	-	-
*R. cyanoxantha*	FH 12-201	Germany	KR364093	KR364225	-	-	-
*R. farinipes*	UE28.09.2002-4	Sweden	DQ421983	-	-	-	-
*R. fuliginosa*	R74	CZECH REPUBLIC	HG798529	-	-	-	-
*R. grisea*	BB 07.184	Slovakia	-	KU237509	KU237795	KU237355	-
*R. grisea*	UE2005.08.16-01	Sweden	DQ422030	DQ422030	-	-	-
*R. grisea*	FH12234	Germany	KT934006	KT933867	-	-	-
*R. ilicis*	MF 00.300	Italy	-	KU237595	KU237880	KU237443	-
*R. illota*	UE26.07.2002-3 (UPS)	Sweden	DQ422024	DQ422024	DQ421967	-	-
*R. indoalba*	AG 15-628	India	KX234820	-	-	-	-
*R. junzifengensis*	HMAS298101^T^	China	OR826832	OR826833	OR915864	OR941507	OR887742
*R. junzifengensis*	HMAS298102	China	OR880061	OR880054	OR915865	OR941508	OR887743
*R. langei*	BB 07.792	France	-	KU237510	KU237796	KU237356	-
*R. livescens*	F0177	China	GU371295	-	-	-	-
*R. maguanensis*	XHW4765	China	MH724918	MH714537	MH939990	-	MH939983
*R. mariae*	BB 07.038	USA	-	KU237538	KU237824	KU237384	-
*R. medullata*	BB 07.252	Slovakia	-	KU237546	KU237832	KU237392	-
*R. mustelina*	FH12226	Germany	-	KT933866	KT933937	-	-
*R. mustelina*	SA 09.88	Slovakia	-	KU237596	KU237881	KU237444	-
*R. nigricans*	-	Germany	AF418607	-	-	-	-
*R. pallescens*	UBC:F23913	Canada	KJ146729	-	-	-	-
*R. pallidula*	RITF2613	China	-	MH027960	MH091698	MW403845	-
*R. pallidula*	RITF3331	China	-	MH027961	MH091699	MW403846	-
*R. parvovirescens*	SDRM 6280	USA	MK532789	-	-	-	-
*R. prasina*	HMAS 281232	China	MH454351	-	-	-	-
*R. pseudocrustosa*	HBAU15015	China	MT337520	-	-	-	-
*R. pulverulenta*	RF566 (pers. herb.)	USA	AY061736	-	-	-	-
*R. rufobasalis*	H17052204	China	MH168570	MK881947.1	-	MK882075.1	MT085585.1
*R. subpunicea*	RITF1435	China: Hunan	-	MW397126	MW411346	MW403839	-
*R. subpunicea*	RITF2615	China: Hunan	-	MW397127	MW411347	MW403840	-
*R. subpunicea*	RITF3715	China	MN833635	MW397124	MW411344	MW403837	-
*R. subpunicea*	RITF2648	China	MN833638	MW397125	MW411345	MW403838	-
*R. substriata*	XHW4766	China	MH724921	MH714540	MH939993	-	MH939986
*R. variata*	BPL241	USA	-	KT933818	KT933889	-	-
*R. vesca*	BPL284	USA	KT933978	KT933839	-	-	-
*R. vesca*	AT2002091	Sweden	DQ422018	DQ422018	DQ421959	-	-
*R. violeipes*	BB 07.273	Slovakia		KU237534	KU237820	KU237380	-
*R. virescens*	HJB9989	Belgium		DQ422014	DQ421955	-	-
*R. xanthovirens*	GDGM 71145	China	MG786056	-	-	-	-
*R. zonatus*	HMAS298099	China	OR826839	OR826846	OR915866	OR941505	OR887740
*R. zonatus*	HMAS298100^T^	China	OR880062	OR880056	OR915867	OR941506	OR887741

Superscript “T” denotes the type strain of the new species.

## Data Availability

All newly generated sequences were deposited in GenBank (https://www.ncbi.nlm.nih.gov/genbank/ (accessed on 10 January 2024)). All new taxa were linked with MycoBank (https://www.mycobank.org/ (accessed on 10 January 2024)).

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
