# Peer review of "Three New Species of Russulaceae (Russulales, Basidiomycota) from Southern China"

_jof, 2024, doi:10.3390/jof10010070_

Round 1

Reviewer 1 Report

Comments and Suggestions for Authors

In the materials and methods, erroneous or in some cases missing information is presented. For example, the primers and amplification methodology for the primer tef are not presented.

The phylogenetic trees need better editing, in the version presented almost no support values are observed.

The descriptions do not present a standardization. Data on the collected materials is missing. In the phylogenetic trees the authors present data for two specimens per described species, but in the descriptions they only name the holotype. What happens with the other collections? More in-depth comparisons with nearby groups and species are missing.

All suggestions, corrections and comments are found in the text.

Comments on the Quality of English Language

There are several errors of misspelled words, errors in punctuation marks. Some paragraphs with very long sentences that lose their meaning. Please review the English for a better understanding.

Author Response

Dear Editors and Reviewers:

Thank you for your letter and comments relating to our manuscript entitled “Three new species of Russulaceae (Russulales, Basidiomycota) from southern China” (ID: jof-2784560). The comments were very helpful in revising and improving our manuscript as well as emphasizing the significance to our research. We have read the comments carefully and made corrections accordingly. Revised portions are marked in blue in the manuscript. The main corrections in the paper and our responses to the reviewer’s comments are given below. We hope that the revisions in the manuscript and our accompanying responses will be sufficient to make our manuscript suitable for publication in the Journal of Fungi.

Responses to the comments of the reviewer:

Reviewer 1#

Comments 1: In the materials and methods, erroneous or in some cases missing information is presented. For example, the primers and amplification methodology for the primer tef are not presented.

Response 1: We have revised the materials and methods section according to the suggestions.

Comments 2: The phylogenetic trees need better editing, in the version presented almost no support values are observed.

Response 2: We have edited the phylogenetic trees and added support values.

Comments 3: The descriptions do not present a standardization. Data on the collected materials is missing. In the phylogenetic trees the authors present data for two specimens per described species, but in the descriptions they only name the holotype. What happens with the other collections? More in-depth comparisons with nearby groups and species are missing.

Response 3: We have rewritten the descriptions, added the data of other collections, and increased the in-depth comparisons with nearby groups and species.

Comments 4: All suggestions, corrections and comments are found in the text. There are several errors of misspelled words, errors in punctuation marks. Some paragraphs with very long sentences that lose their meaning. Please review the English for a better understanding.

Response 4: We have made revisions according to the suggestions and comments in the text, split and rewrote the long sentences, and added electron microscope photos and some descriptions. Thank you very much for your suggestions.

Comments 5: In Line 23-24, I would like to know what these formats are!

Response 5: We have revised the abstract according to the suggestions.

Comments 6: Consider breaking the paragraph into smaller sentences for better readability, especially since it contains a lot of information.

Response 6: We have divided the paragraph into smaller sentences.

Comments 7: In the phrase “inrelatively warm” in line 35, there seems to be a spacing issue. It should be “in relatively warm.”

Response 7: We have changed “inrelatively warm” to “in relatively warm”.

Comments 8: In line 39, consider adding a comma after “Tailao” for better clarity.

Response 8: We have added a comma after “Tailao” according to the suggestion.

Comments 9: In line 41, consider adding a comma after “banyans” for better punctuation.

Response 9: We have added a comma after “banyans” for better punctuation.

Comments 10: In line 46, consider replacing “establishe” with “established.”

Response 10: We have replaced “establishe” with “established”.

Comments 11: In line 99,How was the amplification and the primers used for the tef region?

Response 11: We have added references for the amplification and primers used for the tef region according to the suggestion.

Comments 12: In line 125,At this time it was not yet known that they were new species

Response 12: We have changed expressions in the text.

Comments 13: You should check this number because it does not seem correct. I think you generated much fewer sequences.

Response 13: We have checked and revised the number of sequences.

Comments 14: The trees are in terrible quality and I cannot see the support values, which I suggest placing in a larger size.

Response 14: We have modified the phylogenetic trees.

Comments 15: In line 158, In the phylogenetic analyzes they have two specimens for each described species, however in the descriptions they only present the data of the type species (a collection) and what about the other one that has different GenBank numbers?

Response 15: We have added the data of other collections.

Comments 16: In line 165,How many specimens did you collect?

Response 16: We collected 14 specimens.

Comments 17: In line 165,Compared to who?

Response 17: We compared to that of new species of Russula published by Li et al.(2023) and added the reference. Li, G.J.; Liu, T.Z.; Li, S.M.; Zhao, S.Y.; Niu, C.Y.; Liu, Z.Z.; Xie, X.J.; Zhang, X.; Shi, L.Y.; Guo, Y.B.; et al. Four New Species of Russula subsection Sardoninae from China. J Fungi 2023, 9, 199, doi:10.3390/jof9020199.

Comments 18: In line 165,These measurements are based on how many specimens?

Response 18: Based on the 14 specimens we collected.

Comments 19: In line 197,Why are these measurements not found in the previous species? You must standardize the descriptions.
Response 19: We have standardized the descriptions according to the suggestion.

Comments 20: In line 224, This notation does not seem correct. Use parentheses to indicate which are the extremes.

Response 20: We have made revisions according to the suggestion.

Comments 21: In line 286, With what methodology did you calculate this percentage of similarity?

Response 21: We have changed the description here.

We tried our best to improve the manuscript and made some changes marked in blue in revised paper which will not influence the content and framework of the paper. We appreciate for Editors/Reviewers’ warm work earnestly and hope the revision will meet with your approval. Once again, thank you very much for your comments and suggestions.

Kind regards,

Junzhi Qiu

E-mail address: [email protected]

Reviewer 2 Report

Comments and Suggestions for Authors

Please revise the manuscript

Author Response

Dear Editors and Reviewers:

Thank you for your letter and comments relating to our manuscript entitled “Three new species of Russulaceae (Russulales, Basidiomycota) from southern China” (ID: jof-2784560). The comments were very helpful in revising and improving our manuscript as well as emphasizing the significance to our research. We have read the comments carefully and made corrections accordingly. Revised portions are marked in blue in the manuscript. The main corrections in the paper and our responses to the reviewer’s comments are given below. We hope that the revisions in the manuscript and our accompanying responses will be sufficient to make our manuscript suitable for publication in the Journal of Fungi.

Responses to the comments of the reviewer:

Reviewer 2#

Comments 1: In line 29,indicate authors

Response 1: We have added a new reference here.

Comments 2: In line 175,The ornamentation is not observed, new photos should be taken in Melzer’s reagent.

Response 2: We have added electron microscope photos.

Comments 3: In line 184,What is the mycorrhizal species?

Response 3: We have added the forest type according to the suggestion.

Comments 4: In line 206,repeat photograph does not look good

Response 4: We have replaced it with a clearer photograph.

Comments 5: In line 229,Indicate species.

Response 5: Main tree species in the fagaceous forests include Pinus massoniana, Eurya nitida, Ilex pubescens, etc.

We tried our best to improve the manuscript and made some changes marked in blue in revised paper which will not influence the content and framework of the paper. We appreciate for Editors/Reviewers’ warm work earnestly and hope the revision will meet with your approval. Once again, thank you very much for your comments and suggestions.

Kind regards,

Junzhi Qiu

E-mail address: [email protected]